# I Like This New Me: Unravelling Population Structure of Mediterranean Electric Rays and Taxonomic Uncertainties within Torpediniformes

**DOI:** 10.3390/ani13182899

**Published:** 2023-09-13

**Authors:** Riccardo Melis, Simone Di Crescenzo, Alessia Cariani, Alice Ferrari, Valentina Crobe, Andrea Bellodi, Antonello Mulas, Laura Carugati, Elisabetta Coluccia, Maria Cristina Follesa, Rita Cannas

**Affiliations:** 1Department of Life and Environmental Sciences, University of Cagliari, 09126 Cagliari, Italy; riccardo.melis@unica.it (R.M.); simone.dicrescenzo@unica.it (S.D.C.); abellodi@unica.it (A.B.); amulas@unica.it (A.M.); laura.carugati@unica.it (L.C.); coluccia@unica.it (E.C.); follesac@unica.it (M.C.F.); 2Department of Biological, Geological and Environmental Sciences, University of Bologna, 40126 Bologna, Italy; alessia.cariani@unibo.it (A.C.); alice.ferrari6@unibo.it (A.F.); valentina.crobe2@unibo.it (V.C.)

**Keywords:** cryptic species, electric rays, genetic variability, Mediterranean Sea, taxonomy uncertainties, *Tetronarce nobiliana*, Torpediniformes, *Torpedo marmorata*, *Torpedo torpedo*

## Abstract

**Simple Summary:**

Electric rays are currently poorly investigated; therefore, their biological status, species identification and distribution are often hard to assess. The present work, using mitochondrial sequence data, aimed to investigate (1) the genetic diversity of the three species of electric rays reported in the Mediterranean Sea (*Torpedo torpedo*, *Torpedo marmorata* and *Tetronarce nobiliana*); and (2) the possible occurrence of other hidden species in the area. Our results suggest that (1) the Sardinian seas (Western Mediterranean) host populations characterized by high levels of genetic diversity, significantly different from other areas located in the Eastern Mediterranean basin, deserving special attention; (2) only three species occur in the investigated area; (3) inaccuracies exist in the current taxonomy of the three investigated species, such as the possible occurrence of cryptic species outside the Mediterranean Sea, as well as in other genera/families of the order Torpediniformes. Future studies are needed to address these issues and inform effective conservation measures.

**Abstract:**

The present study focused on the three species of electric rays known to occur in the Mediterranean Sea: *Torpedo torpedo*, *Torpedo marmorata* and *Tetronarce nobiliana*. Correct identification of specimens is needed to properly assess the impact of fisheries on populations and species. Unfortunately, torpedoes share high morphological similarities, boosting episodes of field misidentification. In this context, genetic data was used (1) to identify specimens caught during fishing operations, (2) to measure the diversity among and within these species, and (3) to shed light on the possible occurrence of additional hidden species in the investigated area. New and already published sequences of COI and NADH2 mitochondrial genes were analyzed, both at a small scale along the Sardinian coasts (Western Mediterranean) and at a large scale in the whole Mediterranean Sea. High levels of genetic diversity were found in Sardinian populations, being significantly different from other areas of the Eastern Mediterranean Sea due to the biotic and abiotic factors here discussed. Sardinian torpedoes can hence be indicated as priority populations/areas to be protected within the Mediterranean Sea. Moreover, sequence data confirmed that only the three species occur in the investigated area. The application of several ‘species-delimitation’ methods found evidence of cryptic species in the three species outside the Mediterranean Sea, as well as in other genera/families, suggesting the urgent need for future studies and a comprehensive revision of the order Torpediniformes for its effective conservation.

## 1. Introduction

Elasmobranchs are characterized by an intrinsic and very low variation of morphological characters, able to hinder correct species identification [1,2]. It is for this very reason that traditional morphological approaches may be ineffective in identifying specimens from cryptic, rare, or elusive species [1,3]. In recent years, molecular tools have been proposed to complement the traditional methods; on several occasions, they have helped to fill gaps of knowledge and resolve taxonomic uncertainties [2,4,5,6,7]. Moreover, they have allowed the early detection of invasive or new species or cryptic species [8,9,10] due to their ability to overcome morphological similarities and related specimen misidentifications. Overall, integrative taxonomic methods have become fundamental tools for correct species assignment [1,11].

Electric rays (of the order Torpediniformes) are still poorly investigated, and many aspects of their life history traits remain incomplete [12,13]. Based on both morphological and molecular evidence, Torpediniformes are known to be a monophyletic group, well distinguished from the other rays and skates, and indicated as a sister group of all batoids [14,15,16,17,18]. Despite the robustness of the conclusions regarding the phylogenetic position of Torpediniformes, some uncertainties remain for the systematics of the five families within the order [17,19].

Many Torpediniformes species are classified as Data Deficient (DD), Not Evaluated (NE) or threatened (Vulnerable, VU; Endangered, EN; Critically Endangered, CR) according to the International Union for Conservation of Nature (IUCN) (Appendix A), and therefore acquiring data on their biology, occurrence, distribution, and abundance are of pivotal importance.

Only the family Torpedinidae occurs in the Mediterranean Sea [19,20] with two genera: *Tetronarce* Gill, 1862 and *Torpedo* Dumèril, 1805. These comprise 13 and 12 species, respectively [21], even if undescribed species are known to occur [19,22]. Three electric rays have been reported as native species in the Mediterranean Sea: the ocellate torpedo *Torpedo torpedo* (Linnaeus, 1758), the marbled torpedo *Torpedo marmorata* (Risso, 1810) and the great torpedo ray *Tetronarce nobiliana* (Bonaparte, 1835) [12,19,20]. At least one additional species, *Torpedo sinuspersici* (Olfers, 1831), has been reported in the Eastern Mediterranean Sea [20], suggesting that our knowledge of this group may not be complete. In truth, a few further species have been reported to occur as potential Lessepsian immigrants (i.e., that entered into the Mediterranean Sea through the Suez Canal) [23]. In detail, in Egypt, *Tetronarce tokionis* (Tanaka, 1908) [24], as well as *Torpedo alexandrinsis* (Mazhar, 1987) and *Torpedo fuscomaculata* Peters, 1855, have been reported [25]. Similarly, *T. sinuspersici* has been provisionally listed as a possible Lessepsian immigrant in Syria [26]. These findings, sporadic and geographically restricted, have been considered not well documented, and hence the occurrence of these species has been regarded as doubtful [23]. Apart from Egypt and Syria, none of these species have been retrieved in recent studies, even in other close areas of the Eastern Mediterranean basin (e.g., Greece [27]; Turkey [28]; or in the central and Western Mediterranean basins [20,23,29,30,31,32,33,34,35]).

Despite the very scarce commercial interest, electric rays are frequently caught as bycatch in the Mediterranean waters [12,27,32,34]. Correct identification of specimens is needed to properly assess the impact of fisheries on populations and species. According to Last et al. [19] and Ebert and Nando [20], smooth-edged spiracles are a distinctive characteristic of the genus *Tetronarce*, while spiracles with slender, tentacle- or knob-like papillae on the posterior and lateral margins are present in the genus *Torpedo*. At the same time, the dorsal surface pattern may assist taxonomic identification, resulting in uniform or various light and dark markings in *Tetronarce* and *Torpedo*, respectively. Despite this apparently clear differentiation, Cariani et al. [1] used molecular techniques and reported morphological misidentification between the Mediterranean species, highlighting the occurrence of taxonomic identification issues, especially for juvenile specimens.

Considering that the correct identification of specimens is the first essential step to properly assessing the impact of fisheries on populations and species, the present study aimed to improve the knowledge of the Mediterranean electric rays using genetic data (1) to molecularly identify specimens caught during fishing operations and assess the rate of eventual morphological misidentifications on the field; (2) to measure the diversity among and within these species and characterize the health status of populations; and (3) to shed light on the possible occurrence of additional hidden species in the investigated area or in general in the studied species and other related taxa.

Firstly, sequences for the cytochrome oxidase subunit 1 (COI) and the nicotinamide adenine dehydrogenase subunit 2 (NADH2) mitochondrial genes were newly produced and used to measure the genetic diversity and connectivity for the three native species along the Sardinian coasts (Western Mediterranean Sea). Secondly, new and published sequences were integrated to characterize the genetic diversity of these species at the level of the whole Mediterranean Sea and in nearby Atlantic waters. Finally, given the finding of cryptic species in the three species, the final analyses were performed by retrieving and integrating all sequences available in public databases for Torpediniformes. Several ‘species-delimitation’ methods were also used to check on the possible occurrence of hidden species and taxonomic issues in other species/genera/families within this order.

## 2. Materials and Methods

### 2.1. Sample Collection, Extraction, Amplification and Sequencing

A total of 152 individuals of Torpenidiformes were sampled between 2008 and 2021, within the framework of the International Bottom Trawl Survey in the Mediterranean program (MEDITS; [36]) and the monitoring program of commercial catch (CAMP-BIOL; [37]) off the coasts of Sardinia (Western Mediterranean Sea, General Fisheries Commission for the Mediterranean Geographical Subarea, GFCM-GSA-11, Res. GFCM/33/2009/2). A total of 15 specimens were assigned to *T. nobiliana*, 39 to *T. torpedo* and 98 to *T. marmorata* (Appendix A), following Serena et al. [38]. For each individual, muscle tissue or fin clips were sampled and preserved at −20 °C in 96% ethanol for laboratory analyses. Total genomic DNA (gDNA) was extracted following the Salting Out extraction protocol [39]. PCR reactions were performed in 25 μL total volume containing 1× PCR buffer, 2 mM of MgCl, 0.2 mM of dNTPs, 2.0 pmol of each primer and 0.8 U of DreamTaq DNA polymerase (Thermo Fisher Scientific, Waltham, MA, USA). The amplifications were performed in a Mastercycler EP Gradient S Eppendorf.

A fragment of the mitochondrial gene COI was obtained from each specimen by PCR using the FishF2 and FishR2 primers [40]. Amplifications were performed with an initial denaturation of 2 min at 94 °C, followed by 38 cycles of 30 s at 94 °C, 30 s at 48 °C and 50 s at 72 °C, and a final extension step for 3 min at 72 °C. Similarly, a fragment of the NADH2 gene was amplified for all individuals using the ND2-MetF and ND2-TrpR primers [35]. PCR conditions consisted of an initial denaturation of 3 min at 94 °C, followed by 40 cycles of 30 s at 94 °C, 30 s at 53 °C and 45 s at 72 °C, and a final extension step for 5 min at 72 °C. The quality of extracted gDNA and PCR outcomes were evaluated on 0.8% and 2.0% agarose gel electrophoresis, respectively. All the amplicons were Sanger-sequenced using the same forward primers used during the amplification by the external provider Macrogen Europe (Amsterdam, The Netherlands).

### 2.2. Data Analysis

#### 2.2.1. Genetic Diversity of Sardinian Samples

The new COI and NADH2 sequences obtained were imported in MEGA X [41] and all the sequences, divided per marker, were carefully edited and then aligned with the CLUSTAL W algorithm [42] implemented in MEGA X. The correct amino acidic translation was verified to exclude nuclear mitochondrial pseudogenes [43].

DNASP v.6 [44] was used to (a) estimate the principal indices for mtDNA (the number of haplotypes [H], haplotype diversity [hd], nucleotide diversity [π] and relative standard deviations) for de novo produced sequences and (b) collapse the sequences into haplotypes.

Sequences of COI and NADH2 for the three electric ray species, available on the GenBank and BOLD databases [45,46,47], were downloaded and added to the respective dataset in order to enlarge the analysis to the whole Mediterranean area and adjacent Atlantic Ocean (Appendix A).

To estimate the occurrence of population structuring for the three electric ray species around the Sardinian coasts the Analysis of Molecular Variance (AMOVA; [48]) was performed for COI, and NADH2 using Arlequin v.3.1 [49]. The AMOVA analyses were carried out for each species, both overall and by grouping the samples based on their geographical origin on three hierarchical levels: among geographical areas, among populations within geographical areas and within populations. The significance of fixation indices values was computed by a non-parametric permutation procedure with 10,000 iterations.

#### 2.2.2. Population Structure in the Mediterranean and Atlantic Ocean

The available sequences of COI and NADH2 for the three electric ray species from the GenBank and BOLD databases [45,46,47] were retrieved in order to enrich the analysis by widening the geographical range/scope to the whole Mediterranean area and Atlantic Ocean (Appendix A). The relationships among haplotypes were inferred with the TCS method [50] implemented in the software PopART v.1.7 [51] in order to build a haplotype network.

The software Bayesian Analysis of Population Structure (BAPS) v.6.0 [52] was employed to identify existing differentiated genetic groups (haplogroups or ‘clusters’) within the three species. BAPS was run using the method of “clustering for linked loci”, a codon model with five independent runs and setting the maximum number of clusters (K) to 12. The output of the mixture clustering analysis was used to graphically represent the results using the POPHELPER online tool [53]. The genetic differentiation among ‘clusters’ identified by BAPS was assessed by AMOVA using the same settings previously described.

#### 2.2.3. Species Delimitation within the Order Torpediniformes

Homologous COI and NADH2 sequences were retrieved from public repositories for all species of Torpediniformes. The script-based search of available sequences was performed using a modified version of the Meta-Fish-Lib R scripts proposed by Collins et al. [54]. Apart from a working R installation [55], five executable R scripts were used (available upon request). In addition to R, the following software was used: HMMER v.3.4 [56], RAxML v.8.0.0 [57] and MAFFT v. 7.520 [58]. The R package requirements were managed by *renv* [59]. Duplicates and sequences not voluntarily obtained (e.g., COII or NADH5 sequences) were deleted from the datasets. Details of additional individual sequences included in the analyses are available in Appendix A, respectively.

A molecular taxonomic approach, implementing species delimitation methods, was applied to assess taxonomic uncertainties and accuracy of public data, and unravel the origin of detected errors.

Firstly, to test the accuracy of specimen identification, all sequences were analyzed using a tree-based approach and four methods: (1) the Bayesian tree reconstruction based on the Yule speciation process (YSP, [60]), performed with Beast v.1.10.4 [61]; (2) the Poison Tree Process (PTP, [62]), (3) the Multi-rate Poisson Tree Processes (mPTP, [63]); and (4) the Bayesian Poisson Tree Process (bPTP, [62]).

The YSP analyses were carried out using the GTR+G substitution model (identified as the best model in MEGA), strict clock type, running 50,000,000 MCMC generations sampled every 1000 generations with a 20% burn-in. Log files were analyzed with Tracer v.1.7.2 [64] to evaluate the robustness of the results obtained. The YSP Bayesian trees obtained were summarized in a single Maximum Clade credibility tree using TreeAnnotator included in Beast.

PTP, bPTP and mPTP were carried out using Maximum Likelihood trees (ML), which were obtained in the PhyML v.3.0 web server (http://www.atgc-montpellier.fr/phyml/ (accessed on 26 May 2023) [65] by applying default settings and automatic selection for the best substitution model determined with Akaike Information Criterion (AIC) of the Smart Model Selection (SMS, [66]) already implemented into PhyML v.3.0 as an online tool.

PTP and mPTP analyses were performed on the online software available at http://mptp.h-its.org (accessed on 31 May 2023) using default settings (*p*-value set to 0.001 for PTP). Similarly, the bPTP analysis was conducted on a web server (http://species.h-its.org/ptp/ (accessed on 31 May 2023) set up for 200,000 generations, with a thinning of 100 and burn-in of 0.25. Bayesian trees used as input files of the bPTP analysis were computed by MrBayes v.3.2.7 (major settings: 20,000,000 generations of Markov chain Monte Carlo and four chains, burn-in 0.25; [67]). *Pteroplatytrygon violacea* (Bonaparte, 1832) was employed as outgroups for all analyses (Appendix A).

Secondly, the accuracy of species delineation was also tested using a distance-based approach by performing (a) the Best Close Match Analysis (BCMA, [68]), (b) the Automatic Barcode Gap Discovery (ABGD, [69]) and (c) the Assemble Species by Automatic Partitioning (ASAP, [70]).

The BCMA analysis was carried out in the R package Spider v.1.5 [71], using the *threshOpt()* function to identify the optimal threshold value for our dataset following the author’s tutorial indications [71]. To perform the BCMA, sequences were provisionally attributed to different putative molecular operational taxonomic units (MOTUs) based on their placement in highly supported branches in the YSP trees.

The ABGD method [69] was computed on the online web application (https://bioinfo.mnhn.fr/abi/public/abgd/abgdweb.html (accessed on 31 May 2023). Default values of *p* between 0.001 and 0.1 were used. The number of steps was set to 10. Several values of X from 0.5 to 1.5 were evaluated. Simple distances are reported as these seemed to perform better than the Jukes–Cantor or Kimura 2-parameter method, as also previously suggested in Srivathsan and Meier [72].

The ASAP analysis was carried out using the ASAP webserver [70]; available at https://bioinfo.mnhn.fr/abi/public/asap/asapweb.html (accessed on 31 May 2023), choosing the simple distance (p-distance) to calculate the pairwise distance.

For the COI sequences, MOTUs were also delimited using the Barcode Index Numbers as defined in BOLD (BINs; [46]).

As the different methods can lead to conflicting results, we considered as consensus the results shared by the majority of all methods applied.

## 3. Results

### 3.1. The Three Electric Rays around Sardinian Coasts

DNA was successfully extracted, amplified and sequenced for 130 and 129 individuals for COI and NADH2 mtDNA markers, respectively (Table 1 and Appendix A; Figure 1). A single case of morphological misidentification was detected in the sample TM_111 and the species attribution was corrected from *T. marmorata* to *T. torpedo* before the start of the subsequent analyses.

The new COI sequences (final alignment length 609 bp) showed a total of 109 polymorphic and 104 parsimony sites, counting two, four and seven haplotypes for *T. nobiliana*, *T. torpedo* and *T. marmorata*, respectively (Table 1). The new NADH2 sequences obtained (final alignment length 912 bp) showed 228 polymorphic and 216 parsimony informative sites, counting a total of 7, 5 and 25 haplotypes for *T. nobiliana*, *T. torpedo* and *T. marmorata*, respectively (Table 1). All the haplotype sequences have been deposited in GenBank (COI Accession Numbers: OR536600–OR536612; NADH2 Accession Numbers OR540834–OR540870).

The newly obtained sequences largely increased the number of haplotypes available for the three species. In general, NADH2 sequences were characterized by higher haplotype diversity (overall hd_NADH2_ = 0.898) and higher nucleotide diversity (overall π_NADH2_ = 0.075) than COI sequences (overall hd_COI_ = 0.736; overall π_COI_ = 0.060) (Table 1).

The AMOVA analysis carried out on Sardinian samples allowed us to test different scenarios. In particular, for *T. torpedo* and *T. marmorata*, we clustered the sequences in a single group, two groups (separating the north/western from the south/eastern locations) and three groups (separating north, west and south/east locations) to test for possible differentiation on a geographic basis. Signals of significant genetic differentiation were detected in *T. marmorata* (one group, COI and NADH2) and *T. torpedo* (one group, only COI), but not in the other scenarios investigated (Appendix A). Due to the very few samples available for *T. nobiliana*, the AMOVA for this species was carried out only overall (one group), obtaining no significant values of Φst in all two datasets analyzed (Appendix A).

### 3.2. The Three Electric Rays at a Large Spatial Scale (Mediterranean and Beyond)

The datasets composed by the new sequences obtained during this study were integrated with additional sequences of *T. marmorata*, *T. torpedo* and *T. nobiliana* downloaded from public data repositories, leading to final datasets containing 227 and 145 sequences for COI and NADH2, respectively, covering the whole Mediterranean Sea and the Atlantic and Pacific Oceans. In NADH2, a total of 30 haplotypes were found in *T. marmorata*, while *T. torpedo* and *T. nobiliana* showed only 6 and 10 haplotypes. Similarly, the highest number of haplotypes for COI was found in *T. marmorata*, showing sixteen haplotypes, while only nine and eight were found in *T. torpedo* and *T. nobiliana*, respectively.

In both the COI and NADH2 datasets, *T. marmorata* showed haplotype networks with the most common haplotypes (e.g., Tmar_H2, Tmar_H3) shared among the Sardinian and other Mediterranean samples, as well as the presence of private haplotypes in several Mediterranean areas (Figure 2 and Figure 3; Appendix A). Similarly, *T. torpedo* showed haplotype networks (Figure 2 and Figure 3; Appendix A) characterized by the presence of the most frequent haplotype shared by several specimens from different areas (Ttor_H1), as well as a few less frequent shared and private haplotypes. However, in both species, NADH2 haplotypes of Atlantic origin from Senegal (Tmar_H30 and Ttor_H6; Figure 3) appeared very distant from the Mediterranean sequences. As concerns *T. nobiliana*, the occurrence of divergent sequences from the Western Atlantic was observed (Figure 2: Tnob_H8 from Canada; Figure 3: Tnob_H8 from Rhode Island, USA), as well as the presence of a few private haplotypes (Figure 2). However, the most frequent COI haplotype (Tnob_H1) was shared by several specimens from different areas, even between very distant locations such as the Mediterranean Sea (e.g., Sardinia and Sicily), Atlantic Ocean (Portugal), Australia and New Zealand (Figure 2).

The results of the BAPS analyses confirmed the large distinctiveness of the Senegalese and Western Atlantic sequences, but at the same time they pointed out the possible weak differentiation among areas within the Mediterranean Sea. In the COI datasets of both *T. marmorata* and *T. torpedo*, the BAPS outcomes highlighted the occurrence of at least two clusters (haplogroups) separating the Western and Eastern Mediterranean basins (Appendix A).

The AMOVA analyses confirmed the occurrence of significant differences between the groups delineated by the BAPS outcomes (Appendix A). Significant differences were never found at a local scale (i.e., among Sardinian locations, Appendix A) but only at a larger scale (Appendix A).

In detail, in *T. marmorata*, COI sequences data indicated that the Eastern and Central Mediterranean locations (from Cyprus up to Malta) were significantly different from the Western Mediterranean locations (from Sicily up to the Balearic Islands, including Portugal in the NE Atlantic, Appendix A). Similar results were obtained in *T. torpedo*, where significant differences were found separating the Western Mediterranean (including Portugal) and the Eastern Mediterranean (both including the central Mediterranean locations—Sicily and South Adriatic in the eastern group or placing them in a third separate group, Appendix A).

The NADH2 sequences confirmed the differentiation within the Mediterranean (Western vs. Eastern locations) in *T. marmorata*, but also the clear distinction of Senegal (Central East Atlantic) from the Mediterranean locations both in *T. marmorata* and *T. torpedo* (Appendix A).

As concerns *T. nobiliana*, the AMOVA on a very few sequences allowed us to demonstrate only an overall differentiation (Appendix A).

### 3.3. Electric Rays around the World

Considering that the preliminary results pointed out the occurrence of quite divergent sequences for the three studied species, a more comprehensive analysis was performed to correctly attribute them. To this end, additional sequences were downloaded and added to the respective datasets, both for other close species (within the family Torpedinidae) and distant species (within the order Torpediniformes). The sequence mining led to final datasets containing 468 and 184 sequences for COI and NADH2, respectively, associated with four families out of the five currently described for the order Torpediniformes: Narcinidae, Narkidae, Platyrhinidae and Torpedinidae. When collapsed into haplotypes, the two final datasets contain a total of 153 and 81 haplotypes for COI (34 species) and NADH2 (21 species), respectively.

Considering the results from both delimitation approaches (tree-based and distance-based), a consensus number of 41 and 26 MOTUs were obtained for COI and NADH2, respectively, more than the initially associated species in the public repositories (Figure 4 and Figure 5).

Concerning the Senegalese and Western Atlantic sequences for the three focal species (i.e., *T. torpedo*, *T. marmorata* and *T. nobiliana*), they were recognized as separate MOTUs by most of the methods, and provisionally identified as MOTU 10, *Torpedo bauchotae* Cadenat, Capapé and Desoutter, 1978 and *Tetronarce occidentalis* (Storer, 1843), respectively (Figure 5). Similar outcomes were observed in the family Torpedinidae involving *T. fuscomaculata* and *T. sinuspersici*, whose haplotypes were recognized as separate, forming four distinct MOTUS (i.e., *T. sinuspersici*, MOTU 1, *T. fuscomaculata* and MOTU 2; Figure 4).

On the contrary, the COI and NADH2 data failed to support the validity of the separation between *T. nobiliana* and *Tetronarce macneilli* (Whitley, 1932) or between *T. nobiliana* and *Tetronarce fairchildi* (Hutton, 1872) (Figure 4 and Figure 5).

Likewise, in species included in the families Narkidae and Narcinidae, the number of ‘nominal’ species was not coincident with the number of MOTUs. For instance, two distinct MOTUS were potentially identified in several ‘nominal’ species (Figure 4 and Figure 5: *Narke japonica* (Temminck and Schlegel, 1850), MOTU 5, *Narcine brunnea* (Annandale, 1909), MOTU 6, *Narcine timlei* (Bloch and Schneider, 1801) and MOTU 7, *Narcine maculate* (Shaw, 1804) and MOTUs 8, 9). On the contrary, contrasting results were obtained for *Narcine brasiliensis* (Olfers, 1831) and *Narcine bancroftii* (Griffith and Smith, 1834): they were identified as separated MOTUs only by the NADH2 data (Figure 4 and Figure 5).

At a higher level, our results showed outcomes partially coherent with the accepted taxonomy.

As concerns the family Torpedinidae, the species of the two genera *Torpedo* and *Tetronarce* were clearly separated. However, within the genus *Torpedo*, *T. marmorata* and *Torpedo mackayana* (Metzelaar, 1919) appear distant from the other congeneric species (Figure 4 and Figure 5).

With both datasets, sequences of Narcinidae (genera *Benthobatis* and *Discopyge*) clustered with sequences of Narkidae (genus *Typhonarke*). Moreover, in the family Narcinidae, large genetic distances were computed within the genus *Narcine*, with species assigned to two well-distinct clusters, encompassing species from Indo-Pacific and South American origin in separate clades (Figure 4 and Figure 5).

## 4. Discussion

The present study mainly aimed to deepen our knowledge on the three species of electric rays inhabiting the Mediterranean Sea using molecular tools. This offered the opportunity to acquire new information useful for the management/conservation of the studied species. Later, the collating of all the public sequences for COI and NADH2 available for the order Torpediniformes permitted us to identify new issues or highlight known taxonomic problems that deserve to be addressed as a priority.

### 4.1. The Three Electric Rays around the Sardinian Coasts

Thanks to an intense dedicated sampling realized in the seas around Sardinia (Western Mediterranean), new genetic information was acquired to complement the biological and fishery-related data recently compiled for the same area [12,34].

As concerns the COI gene, the newly generated sequences enlarge the dataset already available for the Mediterranean electric rays [1,35,73,74,75,76], with the addition of seven new haplotypes (Table 1 and Appendix A). Similarly, 21 new NADH2 haplotypes were added to the few available for Mediterranean *T. marmorata*, and for the first time a total of 12 NADH2 haplotypes were obtained from *T. nobiliana* and *T. torpedo* specimens of Mediterranean origin (see Table 1 and Appendix A for details).

Firstly, the application of molecular techniques allowed us to confirm the occurrence of only three distinct species in the Sardinian waters. In addition, the genetic data permitted us to identify errors in the initial morphological attribution; a ray identified as marbled torpedo turned out to be an ocellate torpedo. This was likely due to a mislabeling during the sampling of the tissue. Given the unique color pattern, dorsal coloration and tail and fin proportions of *T. torpedo*, it is difficult to imagine that its clear distinct morphological appearance was not captured during field specimen processing. However, this type of error is not new, as Cariani et al. [1] already described several cases of misidentification involving Mediterranean electric rays. Especially immature individuals of *T. marmorata* are frequently misidentified, due to the lack of clear traits and diagnostic characters in small-sized individuals.

Secondly, we measured moderate-to-high genetic diversities in the Sardinian electric rays (Table 1). High NADH2 haplotype diversity was recorded in Sardinian marbled torpedoes and ocellate torpedoes (Table 1: hd_ND2_ = 0.781 and 0.829 in *T. torpedo* and *T. marmorata*, respectively). From a conservation perspective, this high diversity seems to indicate that the Sardinian electric rays are variable and hence apparently in ‘good health condition’. This potentially enables them to adapt to changes in their environment and/or to stressful conditions such as excessive fishing pressure more efficiently [77]. On the contrary, reduced genetic diversity could have resulted in decreased population viability and increased extinction likelihood [78]. A high genetic variability in Sardinian waters has already been reported for other elasmobranchs [79,80] and attributed to the hydrological conditions and peculiar geographical position of Sardinia. The island, located between the Liguro-Provenzal and Algerian basins, is in a lucky position where the frequent mixing of different gene pools from north and south of the Western Mediterranean basins could generate this variability [81]. These highly variable Sardinian electric ray populations could represent important priority populations and areas for conservation purposes.

Finally, lack of significant genetic structuring was detected for the three species. It is well known that the level of genetic differentiation within marine species results from a complex equilibrium between structuring factors (e.g., oceanographic fronts, isolation by distance) and homogenizing factors (e.g., migratory behavior of adults) [82]. In our case, the high connectivity detected at the small spatial scale along the Sardinian coasts may possibly be explained by the absence of barriers to adult migration leading to unrestricted gene flow between areas. Unfortunately, the results obtained here represent only a first evaluation of the genetic diversity and health status of the three Mediterranean electric rays. Moreover, they suffer from limited marker resolution, as well as the limited number of specimens analyzed, especially for *T. nobiliana* and *T. torpedo*. These low numbers could reflect the rarity of the species at sea and/or the inherent difficulties in sampling them at the markets due to the low economic value and fishery interest.

### 4.2. The Three Electric Rays at a Large Spatial Scale (Mediterranean and Beyond)

Examining the three species of electric rays at a wider geographic scale, the network analysis revealed the occurrence of shared haplotypes among sequences from Mediterranean and extra-Mediterranean specimens, as well the occurrence of private and divergent haplotypes.

The BAPS and AMOVA analyses detected the presence of significant genetic differentiation, even within the Mediterranean Sea between the Eastern and Western locations. This observed genetic structuring could depend both on historical processes (e.g., past geological events), contemporary restrictions to dispersal (e.g., distance, depth) and biological characteristics of the species (e.g., site fidelity, habitat preferences, prey abundance) ([83] and references therein). In particular, the role of oceanographic discontinuities has been invoked in several elasmobranch studies as the main factor responsible for geographically isolating and hence genetically differentiating the populations. This pattern has been reported for the differentiation of the easternmost Mediterranean locations of the Levantine Sea, with the Strait of Sicily potentially representing a barrier limiting the genetic exchanges between the Eastern and the Western Mediterranean basins [1,7,84,85,86,87].

Nevertheless, the results described here are only indicative, based on the available data, opportunistically compiled merging different studies or public sequences. Future studies, based on a more balanced sampling design with adequate numbers of specimens from the different basins of the Mediterranean Sea, are urgently required. Particularly, the easternmost part of the Mediterranean Sea should be investigated with an intensive sampling to shed light on the extent of the genetic differentiation of the populations. Such information is crucial for comprehensive and effective conservation and management of these potentially distinct evolutionary units [88].

As concerns the quite divergent sequences retrieved in public repositories, obtained from specimens caught outside the Mediterranean Sea, they were analyzed using several species delimitation methods, recently applied to other elasmobranch species to address identification issues and solve taxonomic uncertainties [3,6,7,89,90].

Firstly, the NADH2 sequence from specimens of the Central Eastern Atlantic (Senegal), originally deposited under the name *T. marmorata* (JQ518928; [16]), was clearly identified as a different MOTU. It should be possibly attributed to *T. bauchotae*, given its morphological features as described in [16]. Similarly, the NADH2 sequence JQ518930 [16,91], from a Senegalese specimen provisionally identified as *T. torpedo*, was recognized as distinct (MOTU 10) with respect to the ‘true’ Mediterranean ocellate torpedoes, potentially representing a ‘still undescribed’ new species. These results confirm the need to further investigate the Senegalese waters, where additional undescribed elasmobranch species are possibly reported to occur [7]. Finally, sequences from North-Western Atlantic specimens, originally attributed to *T. nobiliana* (NADH2: JQ518931, COI: KC015969), were identified as a different MOTU; they should be attributed to *T. occidentalis* as suggested in [92].

On the contrary, the sequences from South Africa (deposited under the name *T.* cf. *nobiliana*) and sequences from New Zealand/Australia (deposited under the name *T. fairchildi/T. macneilli*) are recognized as being the same MOTUs as the Mediterranean great torpedoes; they are all synonyms of *T. nobiliana* [19].

These scenarios (i.e., mislabeling/misidentification/cryptic speciation) are a common occurrence not only in torpedoes but in chondrichthyan species, making morphology-based identification often unreliable, and highlighting the importance of genetic tools for accurate identification, quantification of catches, delineation of units and definition of the best strategies for fisheries management and conservation [88].

### 4.3. Electric Rays around the World

From our species delimitation analyses, several undescribed species were identified in other areas. For instance, in the Indian Ocean within the family Torpedinidae in the genera *Tetronarce* and *Torpedo* several ‘unidentified’ MOTUs were found (MOTUs 1, 2, 3, and 11; sequences taken from [93,94,95]). In particular, MOTU 1/2, and MOTU 11 were molecularly very close but distinct to *T. sinuspersici* and *T. fuscomaculata*, respectively. These results confirm the occurrence of ‘species complexes’ in both species, showing that vthere are still taxonomic issues to be fully addressed, as suggested by several authors [19,96].

Similarly, in the Indian Ocean, cryptic undescribed species (MOTUs 4, 6, 7, 8, 9) were identified in the families Narkidae and Narcinidae. Most cases are related to sequences deposited in the public repositories under the names of *N. brunnea, N. timlei* and *N. maculata*, but are clearly different from them. Unfortunately, these sequences are often ‘unpublished’ (not included in any peer-reviewed publication), or they are not associated with a detailed description of the voucher specimen. This makes the proper attribution of these sequences to a species hard and ‘questionable’.

On the other hand, new species have been described in the Indian Ocean or reported for the first time in this same area [97,98,99,100]. Rarely have these findings been coupled with molecular analyses to confirm the species attribution. In brief, there are sequences without a proper morphological identification and specimens (species) described morphologically but not molecularly characterized, a ‘common’ practice that should be definitively abandoned.

Finally, a few additional thoughts for future taxonomic studies can be derived from our results related to (1) the genus *Narcine* family Narcinidae and (2) the position of *T. marmorata* within the genus *Torpedo* family Torpedinidae. In all our analyses, sequences of *Narcine* spp. were never grouped together but spread in three–five distinct clusters. Similarly, *T. marmorata* clustered within *Torpedo* but quite apart from other species of this genus. In both cases, the molecular data suggest the possible need for a taxonomic revision, eventually providing for separate subgenera or even distinct genera (to be defined).

As concerns the marker to be used, overlapping results were obtained with the two genes in this study. COI sequences were available for many species, producing a more complete picture of the taxon under study. Conversely, NADH2 sequences were more variable than COI and, in a few cases, outperformed them, such as by being able to distinguish even very close species (i.e., *N. brasiliensis* and *N. bancroftii*). Therefore, it is highly advisable to encourage the use of both markers, in particular to enlarge the number of NADH2 sequences, including underrepresented or lacking species. Moreover, a denser taxon sampling across the species of different genera and families is required to reach robust conclusions.

## 5. Conclusions

The present work highlighted two important needs concerning electric rays: (a) the urgency of quickly improving our knowledge on the distribution and connectivity of the three Mediterranean species and (b) the need for an interdisciplinary action for a reliable taxonomic revision at both the family (Torpedinidae) and order (Torpediniformes) levels for this iconic group.

In brief, our data allowed us to identify Sardinian electric rays as genetically rich, and hence good candidates as priority populations/areas to be monitored through time and eventually protected within the Mediterranean Sea, if changes are recorded (e.g., gene pool rarefaction). Nevertheless, more data are needed to define the degree of differentiation within the species and among the Mediterranean basins. The use of very informative tools (i.e., Single-Nucleotide Polymorphisms, SNPs) could provide new insights into the mechanisms behind the population differentiation, the phylogeographic patterns, the evolutionary history, and the demographic status of the Mediterranean electric rays. Given the difficulties in having a large number of samples to analyze, the use of a large number of genome-wide SNPs could be of help, allowing us to reduce the number of samples required per area [101,102], with a consequent increase in the number of sampling areas analyzed, delving more deeply into the population structure and evolutionary progress at a larger scale.

In recent years, a number of conservation strategies have been implemented or proposed for elasmobranchs, from gear/catch limitations to sanctuaries (discussed in detail in [103]). In particular, MPAs (Marine Protected Areas), restricting some or all fishing activity, have been suggested as valid tools contributing to elasmobranch conservation by limiting mortality and protecting areas of critical habitat within their boundaries [103]. In this context, the present and newly obtained genetic data of electric ray populations could be of extreme importance, as they could inform the design of protected areas and recovery plans, with the correct delineation management units.

In addition, our results highlighted several other areas/species deserving deeper study. Multiple markers (mitochondrial and nuclear genes) or even entire mitogenomes should be used to shed light on the known taxonomic uncertainties, impossible to disentangle with partial, incomplete datasets [104,105,106,107].

## Figures and Tables

**Figure 1 animals-13-02899-f001:**
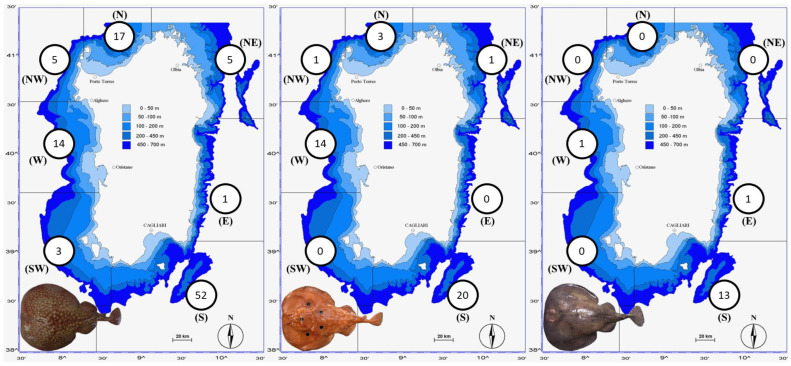
Number of samples sequenced for *T. marmorata* (**left**), *T. torpedo* (**middle**) and *T. nobiliana* (**right**) subdivided in the seven geographical areas delineated in this study: South (S); East (E); Northeast (NE); North (N); Northwest (NW); West (W); Southwest (SW).

**Figure 2 animals-13-02899-f002:**
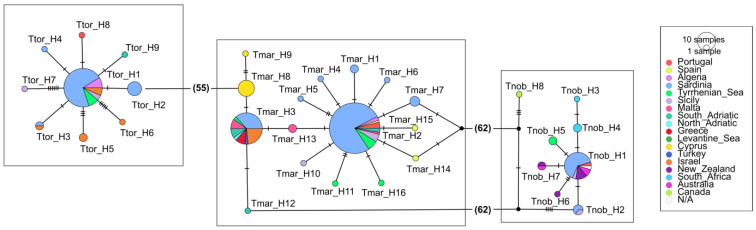
Graphical representation of TCS haplotype network showing *T. torpedo* (**left**), *T. marmorata* (**middle**) and *T. nobiliana* (**right**) for COI. Black dots represent unsampled haplotypes. The numbers between parentheses indicate the number of substitutions separating the two haplotypes.

**Figure 3 animals-13-02899-f003:**
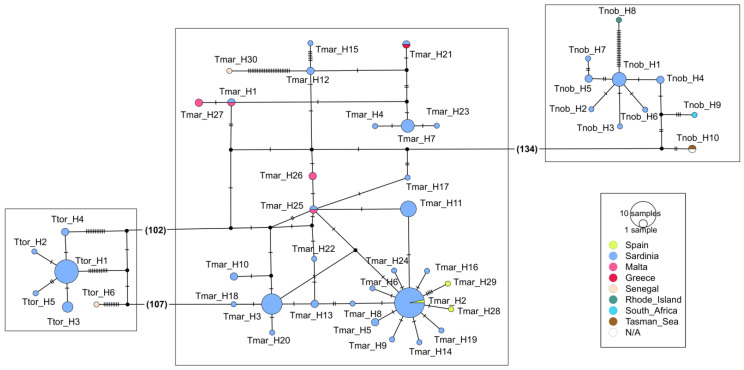
Graphical representation of TCS haplotype network showing *T. torpedo* (**left**), *T. marmorata* (**middle**) and *T. nobiliana* (**right**) for NADH2. Black dots represent unsampled haplotypes. The numbers between parentheses indicate the number of substitutions separating the two haplotypes.

**Figure 4 animals-13-02899-f004:**
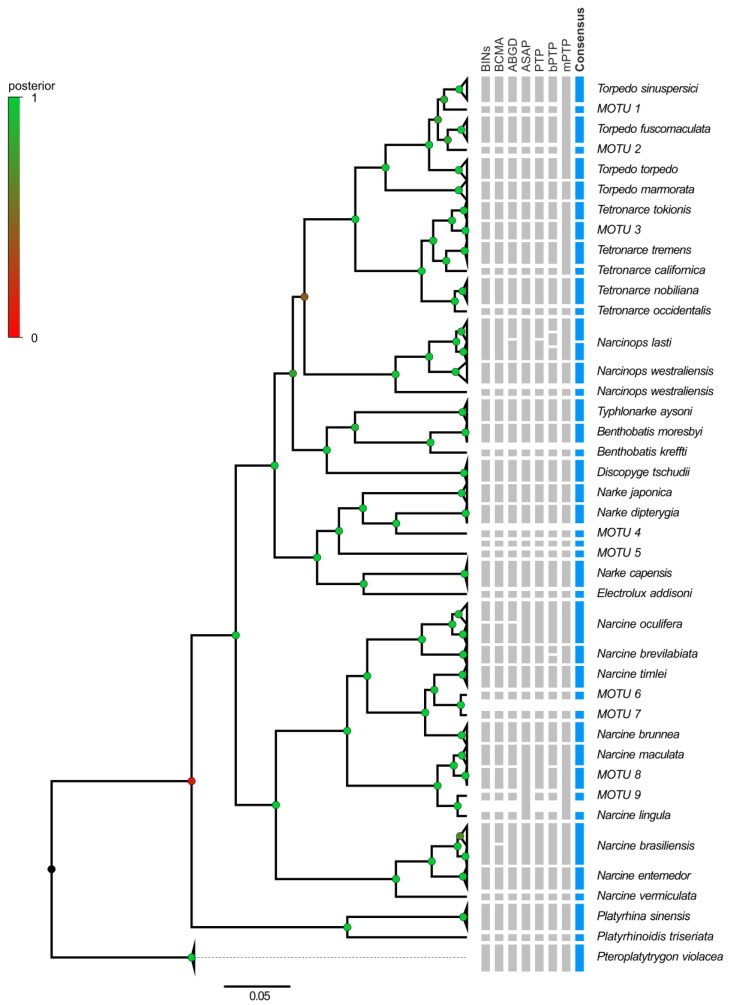
YSP Bayesian tree reconstruction for COI. Posterior probability values at nodes are indicated by the color of circles according to the legend. In the columns on the right, gray bars report each species attribution according to BINs, BCMA, ABGD, ASAP, PTP, bPTP and mPTP. Blue bars represent the consensus taxon according to the seven methods proposed. MOTU refers to taxa not corresponding to a known nominal species.

**Figure 5 animals-13-02899-f005:**
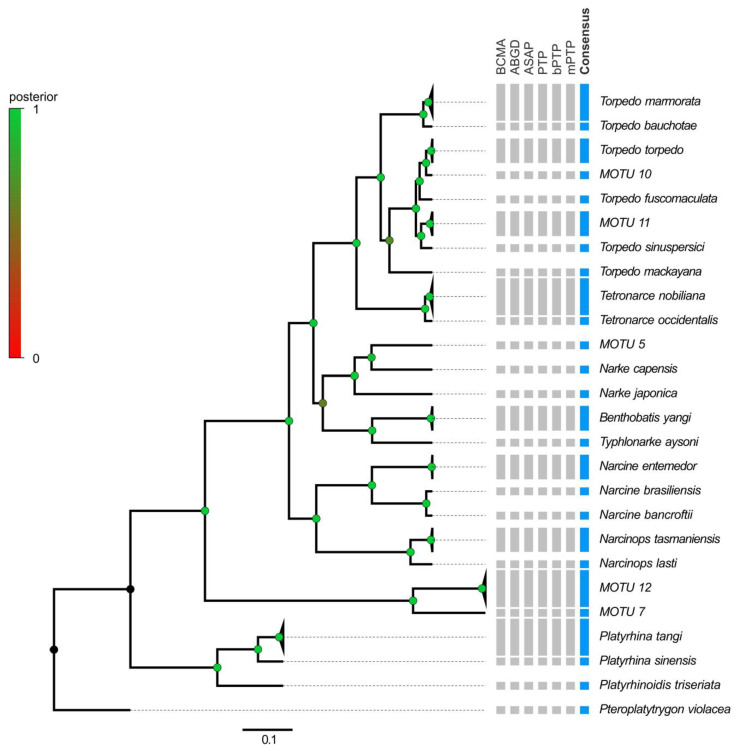
YSP Bayesian tree reconstruction for NADH2. Posterior probability values at nodes are indicated by the color of circles according to the legend. In the columns on the right, gray bars report each species attribution according to BCMA, ABGD, ASAP, PTP, bPTP and mPTP. Blue bars represent the consensus taxon according to the seven methods proposed. MOTU refers to taxa not corresponding to a known nominal species.

**Table 1 animals-13-02899-t001:** Number of sequences (N) in the COI, and NADH2 datasets for the three species analyzed. H = number of haplotypes, with in brackets the number of new haplotypes discovered; hd = haplotype diversity; π = nucleotide diversity.

Species	COI				NADH2			
	N	H	hd	π	N	H	hd	π
*T. marmorata*	77	7 (5)	0.419 ± 0.066	0.001 ± 0.000	86	25 (21)	0.829 ± 0.032	0.004 ± 0.000
*T. torpedo*	38	4 (2)	0.352 ± 0.088	0.001 ± 0.000	28	5 (5)	0.781 ± 0.107	0.001 ± 0.000
*T. nobiliana*	15	2 (0)	0.248 ± 0.131	0.001 ± 0.000	15	7 (7)	0.479 ± 0.016	0.002 ± 0.006
Total	130	13	0.736 ± 0.000	0.060 ± 0.004	129	37	0.898 ± 0.016	0.075 ± 0.006

## Data Availability

All COI and NADH2 haplotypes have been deposited in GenBank (Accession Numbers COI: OR536600–OR536612; NADH2: OR540834–OR540870). Further details are available in Appendix A.

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
