# Peer review of "I Like This New Me: Unravelling Population Structure of Mediterranean Electric Rays and Taxonomic Uncertainties within Torpediniformes"

_animals, 2023, doi:10.3390/ani13182899_

Round 1

Reviewer 1 Report

The authors present an interesting study on genetic differentiation and population structure of Mediterranean electric rays. The bright side of the manuscript is to provide practical details on the current genetic and taxonomic structure of the species. In this context, the study contributes to understanding the evolution of species complex. However, some points are missing (mentioned below) in the manuscript and some parts of the manuscript are not easy to understand. Because of these reasons, major concerns are raised. Therefore, I would like to make some suggestions to improve the quality of the paper as below:

General Comments

Some parts of the manuscript are not easy to understand (mentioned below in specific comments). There are many long sentences and wordiness. This situation disrupts the flow of the subject and the continuity of the reading. Because of this reason, authors should re-reconsider writing some parts of the manuscript.

The Simple summary, Abstract, Introduction and Discussion sections need structural changes. Moreover, it is not strictly needed but the Discussion section can be enriched with a more theoretical interpretation and relate the present results with additional concepts. For instance, the study results can be discussed in the framework of local genetic diversity, gen flow and subpopulation differences in different species from different countries in the broader context.

Specific Comments

Lines 17-31: I think, Simple Summary is too long. Please delete all unnecessary sentences and rephrase the section (Please just add a sentence for each of these: the problem, aim of the study, the most important results and contribution of the study to taxonomy and conservation of the Mediterranean electric rays).

Line 21: Torpedo. marmorata -> Torpedo marmorata

Lines 32-46: In my opinion, the abstract needs to be rephrased and improved.  It is good to start with the problem examined in the study. Within this context, the main problem that is examined by the authors should be explained in 1-3 sentences at the beginning of the abstract. In this context, lack of information about the biology and conservation status and taxonomic uncertainties of Electric rays can be mentioned. After that, the methods and the main results should be given briefly. This can be followed by the main findings of the study. Finally, what is the importance of the results and how the results contribute to further studies should be written down. In my opinion, it is always good to finish the abstract with such a sentence. Furthermore, authors may also say in 1-2 sentences that their findings contribute to further studies and resolving the taxonomic uncertainties of Electric rays. In this way, the bridge between the problem and the solution found by the authors would be stronger.

Line 56:with a globally distribution” -> “, has a global distribution”.

Line 51: Please delete “a taxonomic group”.

Lines 54-57: “In recent years, molecular tools have been proposed to complement the traditional methods; on several occasions, they helped in filling gaps of knowledge and resolve taxonomic uncertainties [2,4-6].” This sentence would better fit here as “In recent years, molecular tools have been proposed to complement the traditional methods; on several occasions, they helped in filling gaps of knowledge and resolve taxonomic uncertainties, species occurrences and genetic contaminations from congenerics [2,4-6].” Please also add these references here (doi: 10.3390/ani13132139, 10.3390/biology12030401).

Line 57: non-yet-described -> new species or cryptic species

Line 59: taxonomy -> taxonomic

Line 63: Torpediniformes results -> Torpediniformes known

Line 68: of the order -> within the order

Line 68: A major part of -> Many

Lines 72-75: “Only the family Torpedinidae occurs in the Mediterranean Sea [18,19]; it encompasses two genera: Tetronarce Gill, 1862 and Torpedo Dumèril, 1805, comprising 13 and 12 species respectively [20], even if undescribed species are known to occur”. Please rephrase here in shorter separate sentences.

Lines 89-98: Please rephrase here and explain the purpose of the study clearly. This part of the paper is important since the authors should explain the purpose of the study and their hypothesis (I mean; what is the problem and what did you do to solve this problem) are given here.

Lines 123-124: Please provide more details for sequencing. For instance “Samples sequenced in both directions on an …….(device name) were automated sequencer by Macrogen (Macrogen Inc., Amsterdam, Netherlands).”

Line 165: Please add the name of the R packages.

Lines 369-375: Please rephrase here in shorter separate sentences since it is not easy to understand.

Lines 381-382: Please explain in more detail. How many new sequences were added?

Lines 385-387: “Once more, the molecular approach has proved his utility for the correct identification of the specimens, allowing us to identify an error in the initial attribution: a single marbled torpedo turned out to be an ocellate torpedo.” Please rephrase here.

Line 385: his -> its

Lines 450-452: “Caution should be applied when interpreting these results due to the small sample size for many localities.” Please rephrase here.

Lines 483-484: “Instead, starting with the three species central to this paper, a few noteworthy points are discussed” Please delete or rephrase this sentence.

Lines 572-582: I think these paragraphs would better fit the Discussion section instead of the conclusion section. 

Some parts of the manuscript are not easy to understand (mentioned below in specific comments). There are many long sentences and wordiness. This situation disrupts the flow of the subject and the continuity of the reading. Because of this reason, authors should re-reconsider writing some parts of the manuscript.

Author Response

We thank very much the reviewer for the constructive comments and suggestions. We did our best to incorporate them in the revised manuscript.

Specific answers to the several points raised are provided in the word file attached

Reviewer 2 Report

The authors of the paper present interesting results of research on selected species from the genera Torpedo and Tetronarce. This work, in comparison with the insufficient number of previous studies, seems to be particularly valuable. The research methods selected for the analyzes are up-to-date and correctly applied, and the description of the results obtained also seems to be carried out correctly.

I have two main comments on the presented manuscript and the first concerns the form of presenting the Discussions. The division of this chapter into three parts (4.1; 4.2; 4.3) seems to be very interesting, and the authors try to refer to the issues raised in a substantive way. However, such a long form of this chapter is not justified in my opinion and I suggest introducing a more synthetic approach, which undoubtedly should have a positive impact on the reception of this valuable work. The same applies to the Conclusions chapter, which is also too extensive and should contain only a few and the most important thoughts.

The second of my comments concerns statements that I feel may be unclear to a potential reader. The authors write "Filling the gaps in knowledge should be prioritized to set proper management plans and effective conservation actions" (page 1, lines 30-31) while in next line (32) we can find "Despite the very scarce commercial interest (.. .)". Therefore, I ask the authors to clarify what kind of management they mean for the species studied.

Other minor comments are listed below:

1. (page1, line 21): please remove the dot in the species name.

2. (page 1, line 24): do the authors deliberately use the plural form of the Sardinian Sea?

3. (Keywords): I suggest alphabetical order.

4. (page 3, line 113): the name of the polymerase should be “DreamTaq” instead of “DeamTaq”.

5. (page 3, line 114): "Mastercicler" appears in the name of the thermal cycler instead of "Mastercycler. Additionally, is the manufacturer of the Eppendorf thermal cycler, as stated by the authors, actually Biometra?

Author Response

We thank very much the reviewer for the constructive comments and suggestions. We did our best to incorporate them in the revised manuscript.

Specific answers to the several points raised are provided in the word file attache

Round 2

Reviewer 1 Report

The authors improved the manuscript wtih the previous comments.